# Regulation Mechanism of MYC Family Transcription Factors in Jasmonic Acid Signalling Pathway on Taxol Biosynthesis

**DOI:** 10.3390/ijms20081843

**Published:** 2019-04-14

**Authors:** Yunpeng Cui, Rongjia Mao, Jing Chen, Zhigang Guo

**Affiliations:** Department of Chemical Engineering, Institute of Biochemical Engineering, Tsinghua University, Beijing 100084, China; yunpengcui@126.com (Y.C.); gowersmao@126.com (R.M.); jingchen.titech@gmail.com (J.C.)

**Keywords:** jasmonate signalling, MYC transcription factors, taxol biosynthesis

## Abstract

Paclitaxel is an important anticancer drug. The phytohormone jasmonic acid can significantly induce the biosynthesis of paclitaxel in *Taxus*, but the molecular mechanism has not yet been resolved. To establish the jasmonic acid signalling pathway of *Taxus media*, based on the gene of the jasmonic acid signalling pathway of *Arabidopsis thaliana*, sequence analysis was performed to isolate the jasmonic acid signal from the transcriptome, a transcriptional cluster of pathway gene homologs and the full length of 22 genes were obtained by RACE PCR at 5′ and 3′: two EI ubiquitin ligase genes, COI1-1 and COI1-2;7 MYC bHLH type transcription factor (MYC2, MYC3, MYC4, JAM1, JAM2, EGL3, TT8); 12 JAZ genes containing the ZIM domain; and MED25, one of the components of the transcriptional complex. The protein interaction between each were confirmed by yeast two hybridization and bimolecular fluorescence complementation based on similar genes interaction in Arabidopsis. A similar jasmonate signaling pathway was illustrated in *T. media*. All known paclitaxel biosynthesis genes promoters were isolated by genome walker PCR. To investigate the jasmonate signaling effect on these genes’ expression, the transcription activity of MYC2, MYC3 and MYC4 on these promoters were examined. There are 12, 10 and 11 paclitaxel biosynthesis genes promoters that could be activated by MYC2, MYC3 and MYC4.

## 1. Introduction

Paclitaxel is an important anticancer drug, which is mainly produced by chemical semi-synthesis by using paclitaxel, which is biosynthesized in Pacific yew. Due to its complex molecular structure, its raw materials have been dependent on *Taxus* biomass, and its biosynthesis and regulation have been studied [1]. The phytohormone jasmonic acid can significantly induce the biosynthesis of paclitaxel in *Taxus*, but the molecular mechanism has not yet been resolved [2,3,4].

The biosynthetic pathway of paclitaxel is very complex, involving at least 19 steps of enzyme-catalysed reactions, and the related enzymes and synthetic pathways have not been fully resolved. The synthetic pathway can be divided into the following four steps: taxane ring synthesis; side hydroxylation, acetylation and phenylacetylation modification; epoxidation modification at C5 position and oxidation at C9; and side chain at C13 position addition [5].

The precursor of taxadiene ring biosynthesis is derived from isopentenyl pyrophosphate (IPP) and dimethyl propylene pyrophosphate (DMAPP), the synthesis, mainly through the plastids of *Taxus* cells [5,6,7]. After ring formation, the biosynthesis process of paclitaxel includes various modifications to the sites on the ring, including hydroxylation, acetylation and benzoylation modification, epoxidation at the C5 position, and oxidative modification at the C9 position [7].

The known modification process is first hydroxylation of the taxane-4(5), 11(12)-diene at the 5α position. This is the rate-limiting step for paclitaxel synthesis [5]. Subsequently, there is hydroxylation of 1β(1βOH), 2α(2αOH), 7β(7βOH), 9α(9αOH), 10β(10βOH), 13α(13αOH), acetylation of 5α(TBT) and 10β(TAT), and phenylacetylation of 2α(DBAT). In addition to the 1β and 9α position hydroxylases, the genes of other enzymes have been confirmed [5]. Oxidation at the 9α position and epoxidation between C4, C5 and C20 occur after the desired modification, resulting in the formation of baccatin III (see Figure 1), but the genes in both steps have not been isolated [5]. At least 11 enzymes are involved in the various modification processes, but the exact order has not been determined. Finally, baccatin III binds to the side chain under the catalysis of the enzyme, and undergoes hydroxylation of the opposite side chain and phenylacetylation to form the final product, paclitaxel. Phenylalanine oxidase (PAM), phenylpropionyl-CoA transferase (BAPT) at position C13, and phenylacetylase (DBTNBT) in the side chain have been identified in this process [5,8].

The biosynthesis step of paclitaxel is very complicated, and it is very difficult to directly increase the expression of the biosynthetic pathway gene to increase the paclitaxel content. As the analysis of all synthetic pathways is not complete, the metabolic pathways cannot be precisely regulated to control the metabolic flux. In this case, for such a complex metabolic pathway, a better choice for regulating genes in the synthetic pathway is to regulate upstream transcription factors, thereby comprehensively regulating the expression of genes in the synthetic pathway [9]. It is important to analyse the transcription factors that regulate the expression of paclitaxel synthesis pathway genes.

Transcription factors regulating Taxol biosynthesis in *Taxus* are being closely studied [2,3,4]. Previous studies have used two methods to isolate related transcription factors: one is to first isolate the promoter of the key gene of the Taxol synthesis pathway, and then to study the transcription factor that can be combined with the promoter; the other is to study jasmonic acid-induced transcription factors in order to analyse whether it can regulate gene expression of paclitaxel.

In the transcription factor-related studies in Taxus, no relevant research results of jasmonic acid signaling pathways in model plants have been used. There is also no systematic and in-depth discussion of the signaling pathway of jasmonic acid in Taxus. Li et al. used high-throughput sequencing to study the transcriptome of jasmonic acid treatment for 16 h, and obtained an AtJAZ-like gene induced by jasmonic acid, and a similar AtMYC2 gene, but no in-depth study [10]. In the transcriptome sequencing study published by the Sun et al., the related gene expression after eight days of jasmonic acid treatment was involved, but no jasmine similar to that in the model plants induced in the early stage of jasmonic acid treatment was involved in the ketoacid signalling pathway transcription factor [11]. Lenka et al. published article on AtMYC2-like genes [8]; except for the TcMYC2 gene, which is known to have a full-length gene, other transcription factors were not found to be similar to AtMYC2 in model plants. The AtJAZ-binding region also does not contain active regions that activate transcription. Maybe due to the shortcomings of the adopted technical methods, the conclusions obtained were also different from those of similar genes in model organisms [12].

In this study, high-throughput sequencing was used to obtain transcription factors after an early induction with jasmonic acid. In our previous study, the sequence of several genes related to the paclitaxel synthesis pathway and jasmonic acid signaling pathway were obtained by sequencing analysis of the transcriptome and expression profile of Taxus. Based on this, the full-length sequence of jasmonic acid signalling pathway-related transcription factors in *T. media* was obtained by sequence comparison analysis and RACE PCR. It includes sequences similar to the important genes of the jasmonic acid signalling pathways such as AtCOI1, AtMYC2, AtJAZ and AtMED25 (At, Arabidopsis thaliana, if the source of the gene is not clearly prefixed, it is from the *T. media*). The interaction between JAZ and MYC transcription factors and other genes was studied, and the jasmonic acid signalling pathway in *Taxus* was confirmed. The above work further studied the regulation of the promoter of the paclitaxel synthesis pathway gene by the MYC transcription factor, and elucidated the regulation mechanism of the transcription factor of the jasmonic acid signaling pathway on the biosynthesis pathway of paclitaxel.

We also found that some of the genes involved in the paclitaxel synthesis pathway were highly expressed by jasmonic acid at 0.5 h, similar to the transcription factors of the jasmonic acid signalling pathway and the expression of self-synthesis-related genes, but faster. Expression of anthocyanin synthesis-related genes regulated by the jasmonic acid signalling pathway was noted at three hours [13]. This indicated that the jasmonic acid signalling pathway plays an important role in regulating the gene expression of the paclitaxel synthesis pathway.

To analyse the mechanism of jasmonic acid signalling pathway specifically regulating the paclitaxel synthesis pathway gene, 14 promoters of paclitaxel synthesis pathway gene were isolated, and five MYC transcription factors were verified by fluorescence activity assay in *Arabidopsis* protoplasts. Regulation of the promoter of the paclitaxel biosynthetic pathway gene and the transcription factor mutants with application value were designed, which lays the foundation for the future construction of a high-yield paclitaxel-based *Taxus* cell line.

## 2. Results

### 2.1. Identification and Analysis of the Full-Length Gene of Jasmonic Acid Signalling Path of Taxus

#### 2.1.1. JAZ

At least eight JAZ-containing ZIM domains were obtained in the differentially expressed genes by the transcriptome sequencing. For comprehensive detection of JAZ that may be contained in the transcriptome, BLAST analysis was performed using the full length of the *Arabidopsis* AtJAZ1 gene and the eight JAZs with reference to the Tm_ZK transcriptome, and the threshold of the E value was set to 0.01. Finally, 16 transcription cluster sequences which may be JAZ were obtained, as shown in Table 1.

For the 16 possible JAZs, specific primers were designed and the full length of the gene was obtained by RACE PCR at the 3′ and 5′ ends. Three of the transcription clusters comp60304_c0, comp64513_c0, and comp68543_c0 failed to obtain a full-length sequence. The other 13 obtained the full length sequence, and wherein comp70749_c0 was a non-coding sequence. The remaining 12 transcription clusters obtained the encoded amino acid sequence—see Table 2.

Homology analysis and domain analysis in the amino acid sequence were performed (see Figure 2a and Figure A1), and the 12 genes were named JAZ1 to JAZ12. 

The primers were designed to amplify the 12 full-length genes, and the cDNA was used as a template for PCR. The bands obtained from the cloning were sequenced to obtain 18 sequences. Among the 18, JAZ1, JAZ2, JAZ9, JAZ11 and JAZ12 all contained varying cleavages. There were three in JAZ2, and the other had two coding sequences (see the Appendix A and Appendix B for the specific sequences). Similar to the variable cleavage of the JAZ gene in model plants, it is important for the diversity of the regulatory functions of the JAZ gene.

#### 2.1.2. COI1

BLAST alignment analysis of the Tm_ZK transcriptome revealed two AtCOI1-like genes, named COI1. To detect the AtCOI1-like gene that may be contained in the transcriptome, BLAST analysis was performed using the full length of *Arabidopsis* AtCOI1 and the above two COI1 with reference to transcriptome, and the threshold of the E value was set to 0.00001. Only the two sequences were obtained, and comp78658_c0 and comp69536_c0 may be the transcription cluster sequence of COI1.

For the two possible COI1, specific primers were designed and the full length was obtained by RACE PCR at the 3′ end and 5′ end. It was found that the two COI1 were actually the full length of the single COI1 gene.

Specific primers to amplify COI1 were produced, and the two highly similar COI1, named COI1-1 and COI1-2, were cloned. Homology analysis was performed with AtCOI1 in *Arabidopsis*, and the results are shown in Figure A2. There were only 10 amino acid differences between COI1-1 and COI1-2, and most of them were not conservative amino acids. Only the amino acid at position 363 was a conservative amino acid variation.

#### 2.1.3. MYC Transcription Factor

The MYC transcription factor belongs to the bHLH type. Ten bHLH-type transcription factors were obtained in the differentially expressed genes by transcriptome sequencing analysis. For comprehensive detection of MYC-like transcription factors that may be contained in the transcriptome, BLAST analysis was performed using the transcription factors of *Arabidopsis* AtMYC2 full length and the 10 bHLH types, and the threshold of the E value was set to 0.01. Finally, 19 possible transcription clusters were found, and homology analysis was performed to obtain full length of RACE for seven transcription clusters of comp78752_c0, comp74670_c0, comp77888_c0, comp77131_c0, comp75263_c0, comp77266_c0, and comp78399_c0. It was found that comp75263_c0 and comp78752_c0 were the sequences at both ends of the same gene (Table 3).

The primers were then designed and cloned using a high-fidelity enzyme to confirm the final gene sequence. The comp77888_c0 was amplified in both cDNA and genomic DNA. In addition to the predicted length bands, a shorter band was obtained. The truncated comp77888_c0 we named comp77888s_c0, so we finally isolated and identified 7 AtMYC2 similar transcription factors in *T.media*: comp74670_c0, comp78752_c0, comp77888_c0, comp77888s_c0, comp77131_c0, comp78399_c0 and comp77266_c0 (see the Appendix A and Appendix B for the sequence).

After homology analysis and domain identification (Figure 2b and Figure A3), the seven transcription factors were named MYC2, MYC3, MYC4, JAM1, JAM2, EGL3 and TT8.

Among them, MYC2, MYC3, MYC4 and JAM1 had higher similarity with AtMYC2 in *Arabidopsis* (Figure 2b), which may be the transcription factor of the core jasmonic acid signalling pathway. The gene expression levels of MYC2, MYC3, and MYC4 were also induced by jasmonic acid, especially MYC3 and MYC4, which were similar to those in the model plants. However, JAM1, which is highly homologous to MYC4, lacked a domain that binds to JAZ and may be a transcription factor that inhibits jasmonic acid signalling.

JAM2 is similar to *Arabidopsis* thaliana’s inhibitory bHLH-type transcription factor AtJAM1, which is also upregulated by the induction of methyl jasmonate treatment, which may be another transcription factor that inhibits the jasmonic acid signalling pathway.

The similarity of transcription factors between TT8 and EGL3 and the jasmonic acid signalling pathway core in model plants was low, and the amino acid sequence of the relevant domain was also different, which is similar to the transcription factor sequence in the regulation of anthocyanin synthesis in model plants. Like the model plants and not induced by the jasmonic acid signal, it is likely to be a transcription factor involved in the regulation of anthocyanin metabolism in *Taxus*.

#### 2.1.4. MED25

In model plants, a transcription factor such as AtMYC2 binds to AtMED25 to recruit RNA polymerase and initiate transcription of a downstream gene of the jasmonic acid signal [14]. At MED25 is an important gene for transcription initiation of jasmonic acid downstream pathway genes and plays an important role in the jasmonic acid signalling pathway. The unique and full length MED25 was obtained in the *Taxus* transcriptome by BLAST analysis. Primers were designed and amplified with high fidelity enzyme to obtain MED25. See the Appendix A and Appendix B for the sequence.

### 2.2. Interaction between Jasmonic Acid Signalling Pathway Genes in Taxus

The jasmonic acid signal is transmitted from the receptor to the protein through the interaction between the proteins. It is necessary to further investigate and detect the jasmonic acid signal by exploring whether the genes of the jasmonic acid signalling pathway are interacting, and to study JAZ and COI1, JAZ and MYC, JAZ and JAZ, MYC and MYC, MYC and MED25 protein interactions, determine whether it is the same as in the model plants, and then construct the jasmonic acid signal in the Taxus.

Since the interaction of the genes has been verified by two methods in model plants, in this study, to detect the interaction between related proteins, the yeast two-hybrid method was used to obtain preliminary interactions between proteins.

#### 2.2.1. JAZ and COI1

Jasmonyl isoleucine can mediate the binding of JAZ to COI1, thereby ubiquitin-labelling JAZ and degrading JAZ-released transcription factors. The combination of AtJAZ and AtCOI1 is an important basis for jasmonic acid signal reception in *Arabidopsis*. This study analysed 12 JAZ and 2 COI1 from the Taxus, and the analog of coronatine, jasmonyl isoleucine. The results are shown in Table 4 (“-”represents no signal, “+” represents a signal, the number of “+”represents the signal’s strength).

In the absence of coronatine, COI1 interacts with JAZ8 and JAZ9. When using 10 μM coronatine, COI1 can interact with JAZ8, JAZ9 and JAZ12 to increase the concentration of coronin to 30 μM. COI1 can be combined with JAZ6, JAZ7, JAZ8, JAZ9, JAZ10 and JAZ12 interaction.

There is also a similar combination of JAZ protein and COI1 protein in the Taxus. However, it is different from the results of the model plant *Arabidopsis*.

#### 2.2.2. JAZ and MYC

In model plants, AtJAZ binds to AtMYCs, thereby inhibiting the transcriptional activity of AtMYCs. In this study, the binding of all JAZ to MYC in *Taxus* was analysed by the yeast two-hybrid method, and the interaction of some JAZ and MYC in tobacco was verified by using bimolecular fluorescence complementation (see Figure 3).

Yeast two-hybrid results showed that transcription factors MYC2, MYC3 and MYC4 interact with most of the JAZ proteins. The exceptions with no interactions are present between MYC2 and JAZ1, MYC3 and JAZ1 and JAZ8, MYC4 and JAZ8. Like the predicted results, JAM1 lacking the JID domain has no interaction with 11 JAZs, but, unexpectedly, it can interact with JAZ8. Another inhibitory transcription factor, JAM2, interacts with JAZ4~JAZ12.

Bimolecular fluorescence complementation verified the interaction between JAZ6 and MYC2, MYC3, MYC4 and JAM2. JAZ also interacts with MYC type and JAM type transcription factors.

#### 2.2.3. JAZ and JAZ

The interaction between AtJAZ and the others is an important condition for the transcription factor inhibition complex, and enriches the means of plant regulation of downstream pathway gene expression through the interaction between differently cleaved JAZ proteins.

The yeast two-hybrid assay was used in this study to analyse the interaction between JAZ in *Taxus* (Figure 4a).

Summarize the interaction between JAZ proteins as shown in Table 5 (“-”represents no signal, “+” represents a signal, the number of “+” represents the signal’s strength):

#### 2.2.4. MYC and MYC

The MYC transcription factor contains the domain of bHLH, in which the domain of HLH is responsible for the binding of MYC transcription factors to each other, forming a homologous or heterologous bimolecular structure in the plant to regulate the transcription of downstream genes. There is no interaction between activated MYC-like transcription factors and inhibitory MYC-like transcription factors in model plants. In this study, the interaction between MYC2, MYC3, MYC4 and JAM1, JAM2 of *Taxus* was analysed. The results are shown in Figure 4b.

Yeast two-hybrid results showed an interaction between MYC2 and MYC3, JAM2 interacted with itself, and there was no interaction between other MYC-type transcription factors.

#### 2.2.5. MYC and MED25

The MYC-like transcription factor binds to MED25, thereby recruiting the RNA polymerase complex and activating transcription. Yeast two-hybrid study of the interaction between MYC2 and other transcription factors in *Taxus* and MED25; the results are shown in Figure 5a.

The activated transcription factors MYC2, MYC3 and MYC4 in *Taxus* can bind to MED25, while JAM1 has no interaction with MED25. Surprisingly, JAM2 binds to MED25, indicating that it may have some transcriptional activation activity. EGL3, which may be involved in anthocyanin synthesis, may also bind to MED25, indicating that it may also have transcriptional activation activity.

### 2.3. Jasmonic Acid Signalling Pathway MYC Transcription Factor Regulates Paclitaxel Biosynthesis

#### 2.3.1. Detection and Analysis of MYC Transcription Factor Activity

To further confirm the activity of the MYC-type transcription factor, the transcriptional activity of the relevant transcription factors was verified in *Arabidopsis* protoplasts (see Figure A4). GAL4BD, which binds to the GAL4 promoter, was fused to a transcription factor and the transcriptional activities of MYC2, MYC3, MYC4, JAM1 and JAM2 in protoplasts were examined.

Among them, the positive control AtMYB21 had strong transcriptional activation activity, and the negative control AtbHLH17 had no transcriptional activation activity. The transcription factor MYC2 in *Taxus* had strong transcriptional activation, and the transcriptional activation activities of MYC3 and MYC4 were low. While JAM1 and JAM2 had almost no activation activity, the results of yeast two-hybridization are consistent with the fact that JAM2 is more active than JAM1.

#### 2.3.2. Synthetic Pathway Gene Promoter Separation and Analysis

To analyse the regulation of transcription factor on the expression of paclitaxel synthesis pathway gene, it is important to isolate the promoter of the synthesis pathway gene. In the existing literature, only a few promoters of the paclitaxel synthesis pathway have been isolated but they were not long enough for the analysis of the promoter properties [2,10].

Therefore, a Genome walker PCR was used to isolate the promoters of all 14 known genes in the Taxol synthesis pathway in *Taxus*. The sequence is shown in the Appendix A and Appendix B. To investigate the possible regulation of these genes by MYC transcription factors, we analysed the number of E-box (CANNTG) that can bind to MYC transcription factors, especially the G-box which has the strongest binding ability to MYC transcription factors. CACGTG sequences and variant G-boxes (AACGTG, CACGAG and special E-box, CATGTG CATGTG) are defined as High Affinity (HA) E-box; AACGTG and CACGAG are defined as modified G-box, and other E-boxes are defined as low affinity (LA) E-box [15]. The number of promoters in the gene was analysed, and the promoter information of the obtained gene is shown in Table 6:

The 14 promoters of th synthetic pathway genes all contain E-box, and all of them are induced by jasmonic acid, and the promoters of six genes contain G-box. The TBT promoter contains the most E-box and the most cis-elements with high affinity for MYC-like transcription factors. The 14βOH promoter contains the least amount of E-box and is low-affinity. The specific location of the cis component is shown in Figure A5.

#### 2.3.3. Activated MYC Transcription Factor Induces a Paclitaxel Biosynthesis Pathway Gene Promoter

Gel electrophoresis migration experiments demonstrated that full-length MYC2 binds to G-box and high-affinity E-box [2]. This indicates that MYC transcription factors have the potential to activate paclitaxel synthesis pathway gene expression. To study the regulation of MYC transcription factors on metabolic pathway genes in this study, plasmids with fluorescent protein and related gene promoter that continuously express transcription factors were transfected into *Arabidopsis* protoplasts. The regulation of MYC2, MYC3 and MYC4 on the promoter of paclitaxel synthesis pathway gene was examined. The principle is shown in Figure A6.

For each promoter, the fluorescence intensity data ratio was normalized to the 62sk plasmid as a reference, and the results were as Figure 6:

The activation of the same transcription factor for the promoters of different genes is shown in Figure 7a–c:

In comparison, among the three activating transcription factors of the jasmonic acid signalling pathway, MYC2, MYC3, and MYC4, although they have different effects on each promoter, MYC2 can activate the expression of all promoter-induced genes. MYC3 and MYC4 had no induction effect on the promoter of BAPT and 14βOH, and both had weaker activation effects on the promoters of TAT and TBT.

The promoters of the genes for the paclitaxel synthesis pathway are substantially similar in relative strength induced by the above transcription factors. The classification is as follows: the promoter of DBTNBT, 2αOH, 7βOH, TS is the strongest; followed by 5αOH, 13αOH, DBAT; followed by PAM, 10βOH, GGPPS; while the promoters of TAT and TBT and BAPT, 14βOH are the weakest. For the promoters of the latter two genes, two of them have an effect of inhibiting transcription.

#### 2.3.4. Inhibitory MYC Transcription Factor Inhibits Activation MYC Transcription Factor

The model plant *Arabidopsis* has four inhibitory MYC transcription factors that bind to the promoter by competitively binding to the activated MYC transcription factor, thereby inhibiting the action of the activated MYC transcription factor [16,17]. For JAM1 and JAM2 isolated from *Taxus*, it was found that there was no corresponding transcriptional activation in protoplasts. The result is shown in Figure 7d,e. In the future, JAM1 and JAM2 overexpression plasmid and reporter genes regulated by all gene promoters should be transferred into protoplasts, and further investigated for it can activate the transcription of the promoter of paclitaxel synthesis pathway gene.

JAM1 and JAM2 had no significant transcriptional activation of all of the tested promoters. Yeast two-hybrid results showed that the inhibitory MYC transcription factor could not bind to the activated MYC transcription factor, similar to that in the model plant. Whether or not it is similar to the model plant, it inhibits the action of the activated transcription factor by competitively binding the promoter sequence to the activated MYC-like transcription factor.

Subsequent studies will co-transfect plasmids expressing inhibitory and activating transcription factors in protoplasts, investigate the strength of transcriptional activity, and analyse whether they have similar inhibitory effects and principles.

#### 2.3.5. Mutant MYC Transcription Factor Activity is not Affected by JAZ

JAZ is an important protein in the signal reception process of jasmonic acid, and it is also a protein that inhibits the activity of transcription factors, and it is also induced by jasmonic acid signal, which plays a feedback inhibition effect in the jasmonic acid signalling pathway. A mutant MYC-like transcription factor in a model plant that does not interact with JAZ but activates transcription of a downstream pathway gene, providing an idea for using biotechnology to increase the signalling pathway of jasmonic acid.

In *Taxus*, JAZ also inhibits the expression of downstream pathway genes by binding to activated transcription factors. This feedback inhibition is an important means by attenuating the effects of JAZ to attenuate the jasmonic acid signaling pathway. However, the number of JAZ in *Taxus* is high, and the way of mutating JAZ to regulate downstream signalling pathways is complicated, and mutations and their interacting MYC transcription factors are feasible methods.

By analysing the complex structure of AtJAZ and AtMYC3 in model plants, the MYC2 and MYC3 variants were designed by MYC2YL, MYC2D, MYC2WDY and MYC3YL, which are the conserved amino acids that play an important role in the binding of JAZ-like transcription factors to JAZ. MYC3D and MYC3WDY (for detailed mutation sites, refer to the sequence of each transcription factor variant in the Appendix A and Appendix B).

First, the binding of the above six MYC transcription factor variants to MED25 was analysed by yeast two-hybrid. The result is shown in Figure 5b. From comparison of results from the unmutated MYC experimental group, it can be seen that, apart from MYC2YL which does not bind to MED25, other mutations do not affect the binding of MYC transcription factors to MED25, indicating that the transcriptional activities of other mutant MYC transcription factors are likely to be unaffected.

The interaction between 12 JAZ and mutant MYC was further studied by yeast two-hybrid assay to analyse whether the mutant MYC could bind to JAZ. The result is shown in Figure 8.

Three forms of mutations all affect the protein interaction between JAZ and MYC transcription factors. Mutation of a single D amino acid site affects the binding of MYC2 and MYC3 to most JAZ, while the mutation of three amino acids of WDY almost prevents MYC2 and MYC3 from binding to all JAZ proteins. The mutation of two amino acids of YL has a stronger effect in MYC2, which makes MYC2 unable to bind to all JAZ, but also cannot bind to MED25; the YL mutation of MYC3 makes it have no interaction with any JAZ except JAZ7.

In summary, the WDY mutation of MYC2 and the YL mutation of MYC3 may be novel transcription factors that are not inhibited by JAZ but have strong transcriptional activity and can be used to improve gene expression of paclitaxel biosynthesis pathway.

The regulation of the MYC-like transcription factor of the mutant on the paclitaxel biosynthetic pathway promoter will be further verified in protoplasts.

## 3. Discussion

### 3.1. Taxus Has a Similar Mechanism of Signal Accepting and Conduction of Jasmonic Acid

As an important phytohormone, jasmonic acid is an important component of plant stress response. It has been proven to be ubiquitous in plants, not only in the model plant *Arabidopsis thaliana* [14,18,19,20,21]. Similar acceptance and transmission pathways for jasmonic acid signals should also be present in the Taxus. Prior to this study, a small number of genes for the jasmonic acid signalling pathway have been isolated by high-throughput sequencing [10], but have not conducted in-depth studies.

The jasmonic acid signalling pathway in *Taxus* was confirmed by high-throughput sequencing and identification of related genes and their function. Twenty-one isolated genes of the jasmonic acid signalling pathways were similar to those found in the model plants, considering characteristic protein domains and protein–protein interactions. It is shown that a similar jasmonic acid signalling pathway may exist in *Taxus* and play an important role.

Firstly, 12 JAZ interacted with two COI1 in the presence of coronatine, and the interaction intensity of some JAZ and COI1 increased with the increase of coronin concentration, indicating that these are probably active molecules of jasmonic acid signalling in *Taxus*. Jasmonyl isoleucine also mediates the interaction between the JAZ and COI1. However, JAZ8 and JAZ9 also have strong interaction with COI1 in the absence of coronatine, which is different from that in model plants. This indicates that the signalling pathways in *Taxus* may be different. The relevant biological significance requires in-depth research.

There is also an interaction between the protein of MYC transcription factor and the JAZ protein in *Taxus*, which is an important way for JAZ to inhibit the transcriptional activities of MYC2, MYC3 and MYC4. JAM1, lacking the JID domain, does not bind to most JAZ, indicating that the mutual binding is based on the JID domain in the MYC-like transcription factor. JAZ8, which contains two EAR domains and has a long length and JAS domain, is not combined with MYC3 and MYC4, but can be combined with MYC2 and the JAM1 without the JID domain. It may play a special role in the jasmonic acid signalling pathway of *Taxus*. Subsequent structural analysis of the complexes formed by AtMYC3 and AtJAZ1 of *A. thaliana* resulted in mutations in MYC2 and MYC3, and mutations in the JID domain resulted in their inability to bind to JAZ. This also suggests that the jasmonic acid signalling pathway in *Taxus* is similar to that in *Arabidopsis* [16,22,23,24]. Interactions exist between JAZ proteins, especially between any combinations of any two of the six JAZs from JAZ7 to JAZ12, but there is less interaction between any combinations of any two of the six JAZs from JAZ1 to JAZ6, indicating that JAZ7–JAZ12 plays an important role in the formation of inhibitory protein complexes [22].

There are also interactions between MYC transcription factors, MYC2 and MYC3, and JAM2 interacts with itself. This differs from the broad interaction between inhibitory MYC transcription factors between activated MYC-like transcription factors in model plants.

The final MYC-type transcription factor recruits RNA polymerase by binding to MED25 to activate transcription of downstream genes. MYC2, MYC3 and MYC4 in Taxus have the same function. JAM1 without activation activity could not bind to MED25. However, JAM2, which has no apparent transcriptional activity, binds to MED25, indicating that it may bind to MED25 competitively, thereby inhibiting jasmonic acid signalling.

### 3.2. Jasmonic Acid Signalling Pathway Transcription Factor Regulates Paclitaxel Biosynthesis Pathway Gene Expression

The transcription factors MYC2, MYC3 and MYC4 of the jasmonic acid signalling pathway play an important role in the paclitaxel biosynthesis pathway. In the promoter region of most paclitaxel synthesis pathway genes, there are G-boxes and similar cis-elements that bind to MYC-type transcription factors, including the promoters of TAT, TBT, DBTNBT, 2αOH, 7βOH, 5αOH, 13αOH, DBAT, 10βOH and PAM, TS. The promoter of BAPT, GGPPS has fewer cis-elements, while the promoter of 14βOH has no corresponding cis-elements.

The activity of the three transcription factors also showed transcriptional activation activity against the promoters of DBTNBT, 2αOH, 7βOH, 5αOH, 13αOH, DBAT, 10βOH, PAM, GGPPS and TS, which contain cis-elements. However, there was no strong transcriptional activation activity for BAPT and 14βOH. It is shown that the G-box cis-element is important for the activity of the above three transcription factors. However, the promoters of TAT and TBT, which contain a large number of cis-like elements similar to G-box, suggest that MYC transcription factors should bind to the transcription factors of the two genes. However, in the *Arabidopsis* protoplasts, they are not strongly activated by MYC transcription factors; in particular, MYC3 and MYC4 have a weak regulation effect on them. Whether the excessive cis-elements in the promoters of TAT and TBT lead to inhibition requires further analysis.

JAM2 has a significant inhibitory effect on all paclitaxel synthesis pathway genes, and its expression is induced by jasmonic acid signalling. It is likely to be a feedback inhibition signal of the jasmonic acid signalling pathway in *Taxus*. The inhibitory effect of JAM1 is weaker than JAM2. JAM2 can be combined with MED25. JAM1 and MYC4 almost have the same sequence. Besides the above functions, whether they are different from model plants, these two transcription factors have other unknown functions and need further research.

In summary, the jasmonic acid signalling pathway in Taxus is summarized in Figure 9. JAR1 catalyzes the binding of JA and isoleucine to form a biologically active jasmonoyl isoleucine (JA-Ile). It can mediate the Coronatine Insensitive 1-E3 ubiquitin ligase complex (SCFCOI1) binding to the jasmonic acid-inducing protein (JAZ) containing the ZIM domain. SCFCOI1 then labels ubiquitin on the JAZ protein. Furthermore, it promotes the degradation of JAZ protein bound to MYC by protease, stops the inhibion of MYC’s activity caused by JAZ’s binding. Eventually, activate the activity of transcription factors such as MYC2, cooperate with different transcription factors, recruit transcriptional complexes including Mediator25 (MED25), and activate genes expression in the downstream pathway. MYC transcription factors play a central role in the *T. media* signaling pathway, and include three MYC factors. The three MYC factors have an upregulation effect on the regulation of genes in each step of the paclitaxel synthesis pathway. In addition to MYC3 and MYC4 having an inhibitory effect of BAPT and 14βOH, JAM1 and JAM2 are MYC transcription factors in *T. media*. They inhibit every step’s gene expression in the downstream pathway.

To confirm the function of the above jasmonic acid signalling pathway-related genes, the construction of related variants is the most important evidence, and the construction of the transgenic method of *Taxus* cell line is a worldwide problem, which is also related to this study. The study will validate the function of related genes and realize the construction of high-yield *Taxus* cell lines in the future.

## 4. Materials and Methods

### 4.1. Cell Culture, Jasmonate Treatment, and RNA Isolation

The *T. media* cell line Zike used in this study was induced from sapling leaf purchased in Beijing. The cells were subcultured using modified SSS medium. For jasmonate treatment, 25 μL 100 mM methyl jasmonate (MeJA) dissolved in ethanol was added to 2 g cells that were suspension cultured in 50 mL of medium 14 days after subculture. Samples were collected after 0, 0.5, 3, and 24 h MeJA treatment using a Büchner funnel. Three biologically independent experiments were performed and a total of 12 samples were obtained. Total RNA was isolated from each of the samples as described before [25]. Generally, the cells were frozen in liquid nitrogen and ground to powder. Every 2 g of cell powder was lysed in 10 mL GTC buffer (62.5 mM Tris-HCl, 12.5 mM EDTA, and 5 M guanidinium isothiocyanate, pH 8.0), then incubated on ice for 15 min and centrifuged at 15,000× *g* for 15 min. The supernatant was added to an equal volume of isopropanol to precipitate the total RNA. The total RNA was purified using an RNeasy Mini kit (Qiagen, City, Germany) according to the manufacturer’s instructions.

### 4.2. RACE PCR

The full length sequence of the 5′ and 3′ ends of the gene was obtained using the GeneRacer Kit with SuperScript III RT kit based on Invitrogen’s RACE PCR technology (CA, USA) using high quality RNA extraction. First, the extracted RNA is dephosphorylated, and then the enzyme is used to remove the 5′ end cap of the RNA, and then the known RNA linker sequence is ligated. Using this as a template, the first strand of the cDNA is synthesized by the oligo(T) used as a locking primer to bind the the 3′ end ploy(A) tail of mRNA. Design specific primers for nested PCR at the 5′ and 3′ ends, respectively, clone the obtained DNA fragments, analyse the sequence of the coding region after sequencing, and after splicing, both stop codons and reach the longest coding region to determine the full length sequence at both ends.

### 4.3. Yeast Two-Hybrid

Based on Clontech’s MATCHMAKER LexA Two-Hybrid System, JAZ and other related genes are ligated into a pLexA vector containing a LexA promoter binding domain (BD), and another gene for detecting interaction is linked to transcriptional activation domain(AD) of the pB42AD vector. The above two plasmids were then transferred to the lacZ-expressing active EGY48 yeast strain containing LexA control using the Frozen-EZ Yeast Transformation Kit of Zymo Research. After overnight culture in SD/–His/–Trp/–Ura liquid medium, transfer 5–20 μL of yeast culture solution to SD/Gal/Raf/–His/–Trp/–Ura/ for activity detection. In X-gal + BU salts solid medium, the appearance of blue spots was observed after four days of culture to evaluate the interaction between lacZ activity and protein.

### 4.4. Bimolecular Fluorescence Complementation

The sequences of the two genes detecting the interaction were ligated into p35SnYFP and p35ScYFP, respectively, and the fusion protein was formed with the C-terminus and the N-terminus of the yellow fluorescent protein YFP, respectively. The assembled plasmids were then transferred to GV3101 *Agrobaterium tumefaciens* by electroporation. After induction with acetosyringone for two days, the two plasmids were simultaneously transferred into the tobacco leaves of the 4-week culture by injection, and then the fluorescence signal of the injection site on the leaves was observed using a laser confocal microscope.

### 4.5. Genome Walker PCR Separation Promoter Sequence

To obtain the promoter of the paclitaxel biosynthetic pathway gene, the genome of *T. media* was extracted using the Plant Genomic DNA Purification Kit (Tiangen, Beijing, China). Based on Clontech’s Universal Genome Walker Kit (Clontech, Takara, Japan), genomic DNA was digested with four different blunt-end restriction enzymes DraI, EcoRV, PvuII, and StuI, followed by ligation to the linker. Primers were designed for two rounds of nested PCR by known coding region sequences and linker sequences, and the obtained DNA bands were cloned, sequenced, spliced and analysed to obtain the promoter sequences of the corresponding genes.

### 4.6. Detection of Transcription Factor Activity in Protoplasts

To analyse the transcriptional activity, the transcription factor was transferred to a plasmid 35SGAL4DB containing a GAL4DB domain in a plant cell to form a plasmid expressing a fusion of the GAL4DB domain and a transcription factor. Based on the mature *Arabidopsis* protoplast transformation method [26], the *Arabidopsis* protoplasts were transferred by a PEG transformation method, and a plasmid containing a GAL4 promoter-controlled transcription factor was simultaneously transformed. The plasmid also contained the reporter gene for the LUC fluorescent protein in the plasmid and the plasmid for the continuous expression of the REN fluorescent protein internal reference. The fluorescence intensity changes of the two fluorescent proteins were detected using the Promega Dual Luciferase Reporter Assay system kit (Cat. E1910) (Madison, WI, USA), and the activity of the transcription factors in the plant cells was analysed by comparing the changes in the fluorescence intensity ratio.

### 4.7. Detection of Transcription Factor Regulation in Protoplasts

The promoter of the paclitaxel biosynthetic pathway gene was inserted into the coding region of the LUC fluorescent protein in the 0800 plasmid to construct a plasmid, which contained the reporter gene with promoter-controlled LUC fluorescent protein and the continuous expression of REN fluorescent protein as internal reference. At the same time, the transcription factor to be detected is inserted into another plasmid 62sk, and the transcription factor can be continuously expressed in the plant by 35S regulation. The two plasmids were co-transformed into protoplasts by PEG transformation. The fluorescence intensity of the two fluorescent proteins was simultaneously detected using the Promega Dual Luciferase Reporter Assay system (Cat. E1910) kit, and the regulation of the transcription factor on the paclitaxel biosynthesis pathway promoter was analyzed.

## 5. Conclusions

In this study, based on the jasmonic acid signalling pathway of the *Arabidopsis* model, the jasmonic acid signalling pathway of *T. media* was isolated and identified. The interaction between related proteins was analysed, and the molecular mechanism of the jasmonic acid signalling pathway in *Taxus* was constructed. Further analysis of the activation or inhibition of the promoter of paclitaxel biosynthesis pathway gene by transcription factors such as MYC2 confirmed the regulation of MYC2 and other transcription factors on the biosynthesis pathway of paclitaxel. Based on this design, the transcriptional activity of MYC2 and MYC3 was not affected by the JAZ inhibition function, which laid a foundation for the construction of a high-yield paclitaxel variant *Taxus* cell line.

*Taxus* has a corresponding jasmonic acid signalling pathway, and it directly regulates the expression of paclitaxel biosynthesis pathway genes through MYC transcription factors, and JAZ and non-transcriptionally active MYC transcription factors comprehensively regulate the expression of related genes. Based on the signalling pathway, a variant MYC transcription factor was constructed to activate transcription without inhibition by JAZ protein, and was the basis for constructing a taxol-producing *Taxus* cell line.

## Figures and Tables

**Figure 1 ijms-20-01843-f001:**
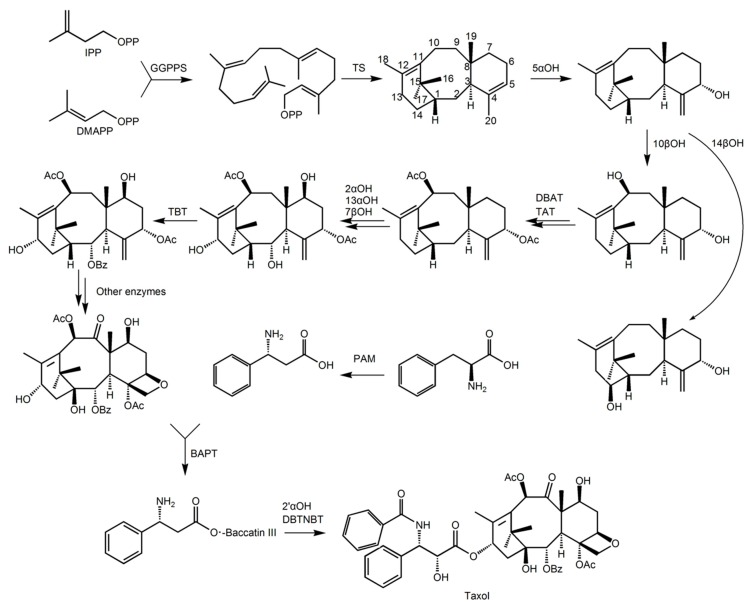
Schematic diagram of the main steps of the paclitaxel biosynthesis pathway. The short list is as follows: GGPPS, di-geranyl pyrophosphate synthase; TS, taxadiene synthase; 5αOH, taxadiene 5α hydroxylase; 10βOH, Taxol 10β hydroxylase; 14βOH, violet Cedar 14β hydroxylase; TAT, taxadiene-5α-ol-acetyltransferase; DBTNBT, 3′-N-dephenylated paclitaxel-N-benzoyltransferase; TBT, taxane 2 α-O-phenylacetyltransferase; DBAT, 10-deacetylbaccatin III-10-O-acetyltransferase; 13αOH, taxane 13α hydroxylase; BAPT, phenylpropionyltransferase; 2αOH, Taxol 2α hydroxylase; 7βOH, paclitaxel 7β hydroxylase; PAM, phenylalanine isomerase; 2′αOH, paclitaxel 2′α hydroxylase.

**Figure 2 ijms-20-01843-f002:**
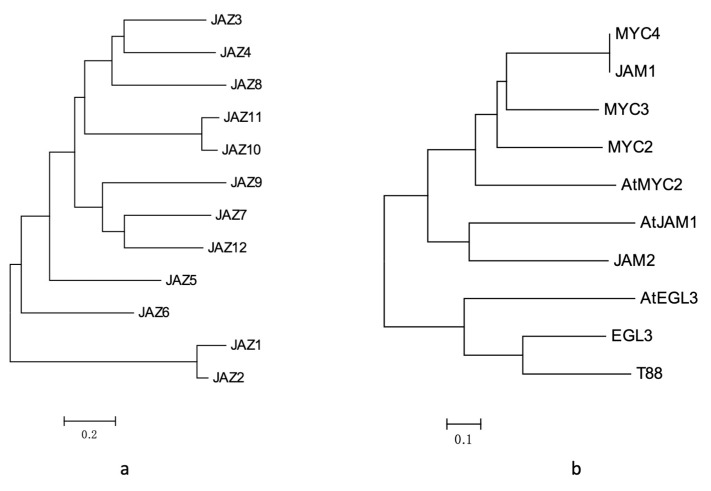
(**a**) analysis of JAZ gene evolutionary tree in *Taxus*; (**b**) results of phylogenetic tree analysis of MYC transcription factors in *Taxus*.

**Figure 3 ijms-20-01843-f003:**
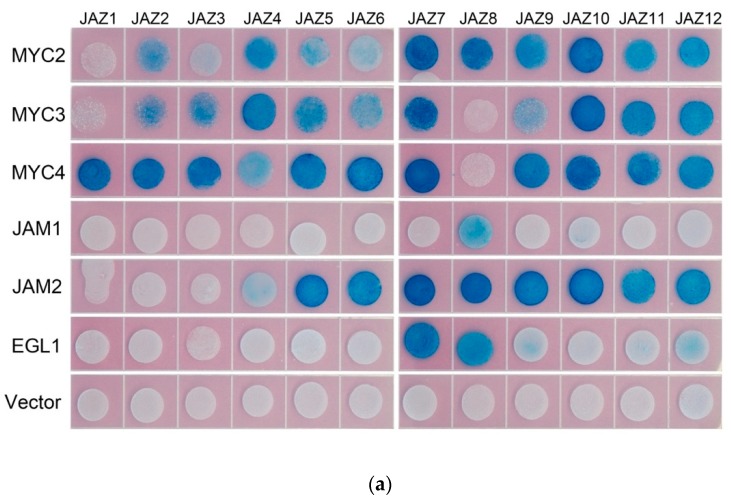
(**a**) yeast two-hybrid analysis of the interaction between JAZ and MYC2-like transcription factors. JAZ was ligated into pLexA and MYCs were ligated into pB42AD for yeast two-hybrid analysis.; (**b**) bimolecular fluorescence complementation analysis of the interaction between JAZ and MYCs.

**Figure 4 ijms-20-01843-f004:**
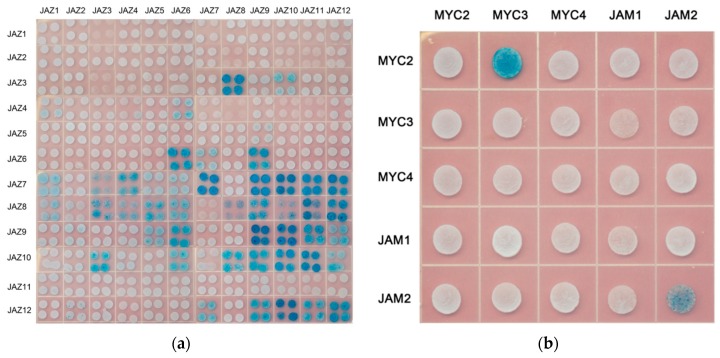
Yeast two-hybrid analysis (**a**) protein interactions between JAZ and JAZ; (**b**) protein interactions between MYC-type transcription factors.

**Figure 5 ijms-20-01843-f005:**
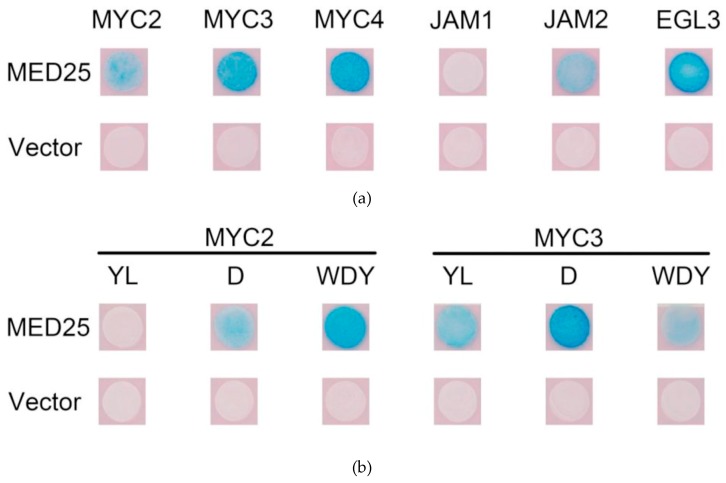
(**a**) Yeast two-hybrid analysis of protein interactions between MYC-type transcription factors and MED25; (**b**) results of yeast two-hybrid assay for mutated MYC-type transcription factors interacting with MED25.

**Figure 6 ijms-20-01843-f006:**
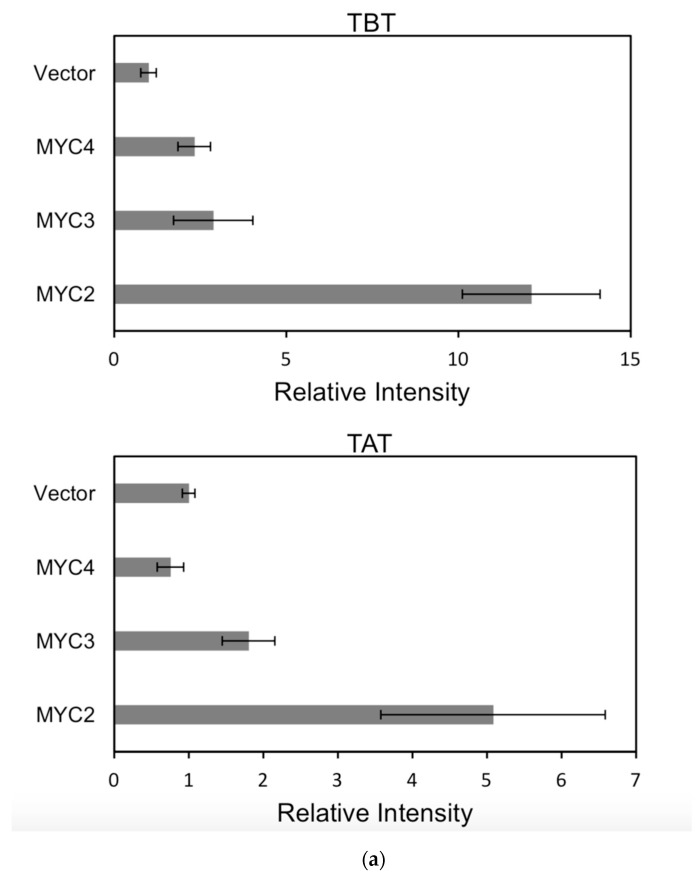
(**a**) transcriptional activity of MYC2, MYC3 and MYC4 on the paclitaxel synthesis pathway genes TAT and TBT promoters (Results are presented as a LUC/REN ratio. *n* = 3 in all samples, error bars are SE(TBT group, * *p* = 3.26 × 10^−6^; TAT group, * *p* = 7.26 × 10^−6^); (**b**) transcriptional activity of MYC2, MYC3 and MYC4 on the paclitaxel synthesis pathway genes DBTNBT, 2αOH and 7βOH promoters (Results are presented as a LUC/REN ratio. *n* = 3 in all samples, error bars are SE(DBTNBT group, * *p* = 9.19 × 10^−8^; 2αOH group, * *p* = 3.02 × 10^−7^; 7βOH group, * *p* = 4.95 × 10^−8^); (**c**) transcriptional activities of MYC2, MYC3 and MYC4 on the paclitaxel synthesis pathway genes 10βOH, PAM and TS promoters (Results are presented as a LUC/REN ratio; *n* = 3 in all samples, error bars are SE (10βOH group, * *p* = 3.86 × 10^−4^; PAM group, * *p* = 0.0264; TS group, * *p* = 0.00528 ); (**d**) transcriptional activities of MYC2, MYC3 and MYC4 on the paclitaxel synthesis pathway genes 5αOH, 13αOH and DBAT promoters (Results are presented as a LUC/REN ratio. *n* = 3 in all samples, error bars are SE(5αOH group, * *p* = 3.21 × 10^−6^; 13αOH group, * *p* = 4.67 × 10^−11^; DBAT group, * *p* = 0.00359 ); (**e**) transcriptional activity of MYC2, MYC3 and MYC4 on paclitaxel synthesis pathway genes BAPT, GGPPS and 14βOH promoters (Results are presented as a LUC/REN ratio. *n* = 3 in all samples, error bars are SE(BAPT group, * *p* = 1.31 × 10^−6^; GGPPS group, * *p* = 3.37 × 10^−5^; 14βOH group, * *p* = 3.72 × 10^−7^).

**Figure 7 ijms-20-01843-f007:**
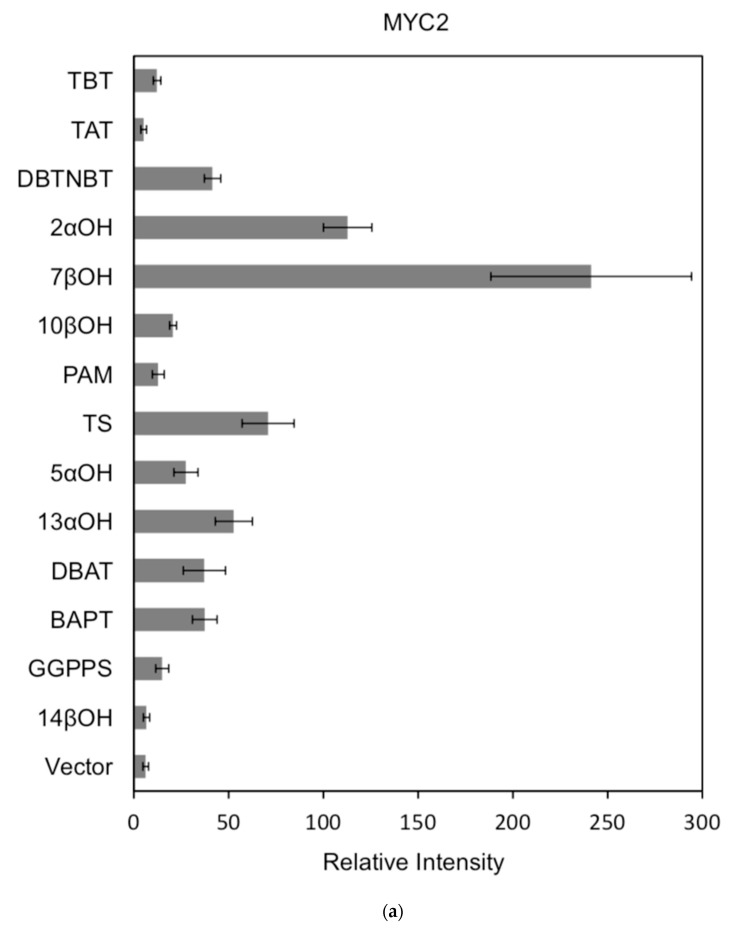
(**a**) transcriptional activity of MYC2 on the promoters of 14 genes known for the paclitaxel synthesis pathway (Results are presented as a LUC/REN ratio. *n* = 3 in all samples, error bars are SE (* *p* = 2 × 10^−5^); (**b**) transcriptional activity of MYC3 on the promoters of 14 genes with known paclitaxel synthesis pathways (Results are presented as a LUC/REN ratio. *n* = 3 in all samples, error bars are SE (* *p* = 1.15 × 10^−11^); (**c**) MYC4 pairs Transcriptional activity of promoters of 14 genes known for the paclitaxel synthesis pathway (Results are presented as a LUC/REN ratio. *n* = 3 in all samples, error bars are SE (* *p* = 2.43 × 10^−6^); (**d**) transcriptional activity of the promoter of the paclitaxel synthesis pathway partial gene (Results are presented as a LUC/REN ratio. *n* = 3 in all samples, error bars are SE (* *p* = 1.7 × 10^−7^); (**e**) transcriptional activity of the promoter of the paclitaxel synthesis pathway partial gene (Results are presented as a LUC/REN ratio. *n* = 3 in all samples, error bars are SE (* *p* = 1.71 × 10^−7^).

**Figure 8 ijms-20-01843-f008:**
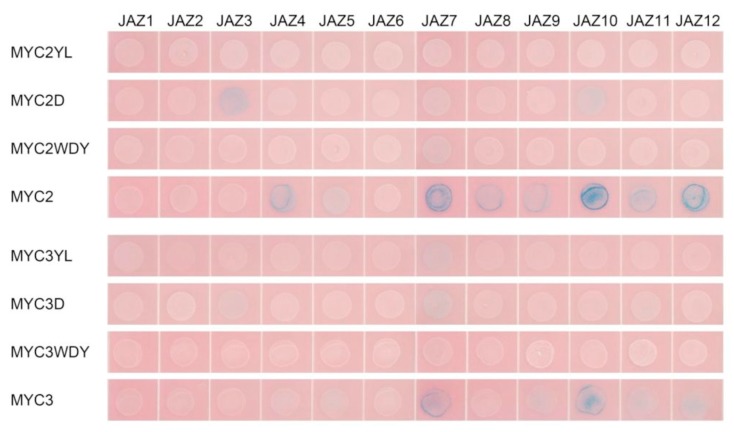
Results of the interaction between the MYC-type transcription factor and JAZ in a yeast two-hybrid assay.

**Figure 9 ijms-20-01843-f009:**
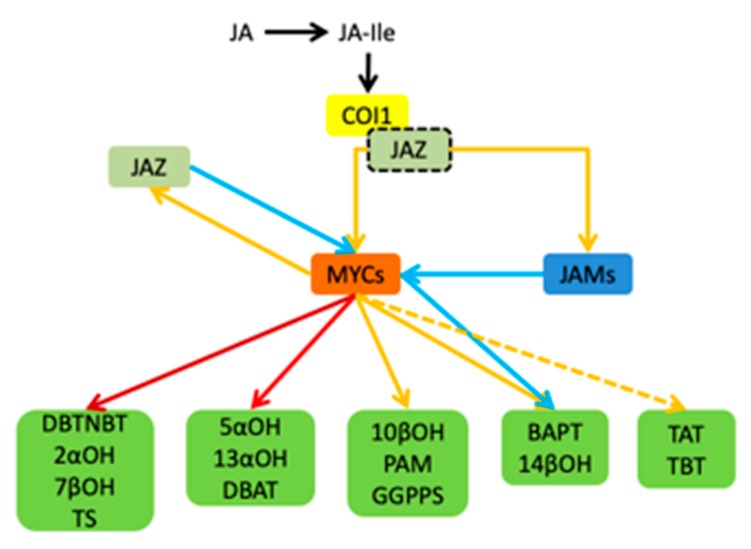
Schematic diagram of the jasmonic acid signalling pathway of *T. media* and its regulation of the paclitaxel biosynthesis pathway gene by the MYC transcription factor. The yellow solid line indicates that the upregulation or regulation is activated, the red solid line indicates a large upregulation, and the dark red solid line indicates a very large upregulation. The yellow dotted line indicates that there may be regulation, and the blue solid line indicates inhibition.

**Table 1 ijms-20-01843-t001:** List of suspected JAZ genes in the Tm_ZK transcriptome.

Unigene ID	Hit Gene Description	Similarity (%)	RPKM			
			0 h	0.5 h	3 h	24 h
comp47611_c0	-	-	11.1	18.8	99.8	135.7
comp60304_c0	-	-	0.2	0.2	2.9	1.5
comp64513_c0	-	-	0.3	1.2	51.4	41.1
comp65899_c0	Protein TIFY 9;	41.94	0.9	9.1	131.5	81.3
comp66635_c0	Protein TIFY 9;	40.66	0.1	1.2	25.5	2.9
comp66717_c0	Protein TIFY 9;	32.52	40.4	92.2	419.4	220.0
comp68543_c0	-	-	0.4	0.9	27.3	1.0
comp70749_c0	-	-	0.1	10.8	97.8	2.5
comp73423_c0	Protein TIFY 6B;	28.37	45.1	42.1	49.0	63.0
comp73780_c0	Protein TIFY 9;	42.11	17.6	42.9	177.5	85.9
comp73885_c0	Protein TIFY 10B;	41.03	20.5	28.8	111.3	116.0
comp75328_c0	-	-	0.3	14.0	101.3	2.7
comp76274_c0	Protein TIFY 9;	40.22	2.6	10.1	122.8	11.1
comp76841_c0	Protein TIFY 6B;	30.12	36.9	49.0	140.3	71.3
comp77597_c0	Protein TIFY 6B;	35	0.2	3.1	34.2	2.3
comp78571_c0	Protein TIFY 6B;	33.01	0.1	2.8	24.4	0.6

**Table 2 ijms-20-01843-t002:** List of full-length JAZ genes isolated from *Taxus*.

Unigene ID	Name	Length	Folds		
			0.5 h	3 h	24 h
comp47611_c0	JAZ11	171	1.7	9.0	12.2
comp65899_c0	JAZ7	142	10.1	146.1	90.3
comp66635_c0	JAZ12	200	12.0	255.0	29.0
comp66717_c0	JAZ6	205	2.3	10.4	5.4
comp73423_c0	JAZ4	393	0.9	1.1	1.4
comp73780_c0	JAZ5	182	2.4	10.1	4.9
comp73885_c0	JAZ10	168	1.4	5.4	5.7
comp75328_c0	JAZ1	225	46.7	337.7	9.0
comp76274_c0	JAZ9	202	3.9	47.2	4.3
comp76841_c0	JAZ3	448	1.3	3.8	1.9
comp77597_c0	JAZ8	530	15.5	171.0	11.5
comp78571_c0	JAZ2	201	28.0	244.0	6.0

**Table 3 ijms-20-01843-t003:** List of MYC transcription factors in taxus.

Unigene ID	Name	AA Length	Folds		
			0.5 h	3 h	24 h
comp74670_c0	MYC2	662	1.35	1.72	1.05
comp77131_c0	JAM2	587	1.54	4.63	1.64
comp78752_c0	MYC3	843	1.58	3.33	1.63
comp77888_c0	MYC4	634	1.78	3.44	1.91
comp78399_c0	EGL3	686	0.92	0.96	1.24
comp77266_c0	TT8	734	0.96	1.71	1.04
comp77888s_c0	JAM1	566	-	-	-

**Table 4 ijms-20-01843-t004:** Yeast two-hybrid analysis. The results of the interaction between JAZ and COI1.

Cor (μM)	BD	pB42AD, Prey(AD)											
		JAZ1	JAZ2	JAZ3	JAZ4	JAZ5	JAZ6	JAZ7	JAZ8	JAZ9	JAZ10	JAZ11	JAZ12
0	COI1-1	-	-	-	-	-	-	-	++	-	-	-	-
	COI1-2	-	-	-	-	-	-	-	-	++	-	-	-
10	COI1-1	-	-	-	-	-	-	-	++	-	-	-	-
	COI1-2	-	-	-	-	-	-	-	-	++	-	-	+
30	COI1-1	-	-	-	-	-	+	-	++	+	-	-	+
	COI1-2	-	-	-	-	-	-	+	+	++	+	-	++

**Table 5 ijms-20-01843-t005:** Yeast two-hybrid analysis of protein interactions between JAZ.

		pB42AD, Prey(AD)											
		JAZ1	JAZ2	JAZ3	JAZ4	JAZ5	JAZ6	JAZ7	JAZ8	JAZ9	JAZ10	JAZ11	JAZ12
pLexA Bait(BD)	JAZ1	-	-	-	-	-	-	-	-	-	-	-	-
	JAZ2	-	-	-	-	-	-	-	-	-	-	-	-
	JAZ3	-	-	-	-	-	-	-	+++	-	++	-	-
	JAZ4	+	-	-	+	-	++	-	-	-	-	-	-
	JAZ5	-	-	-	-	-	-	-	-	+	-	-	-
	JAZ6	-	-	-	-	-	+++	++	-	++	-	-	-
	JAZ7	++	-	+	++	+	+	++++	-	+++	+++	+++	+++
	JAZ8	+	-	++	+	++	++	-	++	+++	++	+++	+++
	JAZ9	-	+	-	-	++	++	-	-	++++	++++	+++	+++
	JAZ10	-	-	+++	-	-	+++	-	+++	+++	+++	+++	++
	JAZ11	-	-	-	-	-	-	-	-	-	-	-	-
	JAZ12	-	-	-	-	-	-	+++	-	+++	++++	+++	+++

**Table 6 ijms-20-01843-t006:** List of cis-element analysis information in the paclitaxel biosynthesis pathway gene promoter sequence.

Gene Name	Promoter Length	G-Box	Modified G-Box	HA E-Box	Total HA cis-Element	LA E-box	Folds at 3 h
TBT	1378	0	2	7	9	5	94.1
TAT	1153	1	4	2	7	4	39.9
DBTNBT	1257	1	2	2	5	4	9.9
2αOH	1179	1	0	3	4	3	16464.7
7βOH	1481	0	0	4	4	2	748.5
10βOH	1144	0	1	3	4	4	14.8
PAM	501	1	1	2	4	2	656.5
TS	1183	1	3	0	4	5	16.3
5αOH	1002	2	0	1	3	3	33.3
13αOH	734	0	2	1	3	3	3155.3
DBAT	1820	0	1	1	2	6	48
BAPT	976	0	1	0	1	2	19.5
GGPPS	970	0	1	0	1	6	3.5
14βOH	792	0	0	0	0	1	33

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
