# Peer review of "Regulation Mechanism of MYC Family Transcription Factors in Jasmonic Acid Signalling Pathway on Taxol Biosynthesis"

_ijms, 2019, doi:10.3390/ijms20081843_

Round 1
Reviewer 1 Report
Dear Authors and Editor,
Although previously studies showed that MYC transcription factor are involved in regulation of jasmonic acid mediated paclitaxel biosynthesis, this regulation is not completely elucidate. This paper entitled “Regulation mechanism of MYC family transcription factors in jasmonic acid signalling pathway on Taxol biosynthesis” give us more thoroughly insight in these regulation mechanisms in order to enable development of efficient strategies for taxol production. Therefore, this study contributes to a better knowledge on the regulation of paclitaxel biosynthetic pathways, and leads to identification of target genes/molecules for the metabolic engineering.
Before being acceptable for publication in “International Journal of Molecular Sciences”, manuscript needs to be language edited more thoroughly, because there are some long, confusing and incomplete sentences.
All comments and suggestions for changes are given in attached document.

Author Response
Thank you for your kind comments on ourmanuscript entitled “Regulation mechanism of MYC family transcription factors in jasmonic acid signalling pathway on Taxol biosynthesis”. We have carefully revised the manuscript according to the reviewer’s comments. Based on the suggestions, we have made an extensive modification on the revised manuscript. Detailed revision was shown in attached documents. The changes to our manuscript were also shown in the revised vision. Thank you very much.
Reviewer 2 Report
In this manuscript, authors have isolated and identified the genes responsible for jasmonic acid signalling pathway of yews from the transcriptome. They identified EI ubiquitin ligase genes, MYC transcription factors, JAZ genes containing the ZIM domain and MED25. Authors studied the interactions between COI1 and JAZ; MYC transcription factor and JAZ; JAZ and JAZ; MYC and MYC; MYC and MED25. The transcriptional activity of MYC2 on the promoters of different genes known for the paclitaxel synthesis pathway was analysed. As jasmonic acid can significantly induce the biosynthesis of paclitaxel in Taxus, understanding the regulation of transcriptional activity is necessary to build a foundation for the construction of a high-yield paclitaxel variant Taxus cell line. Although a lot of information is here, authors should present the data in a systematic way to execute a proper model.
Abstract
The lines are lengthy and difficult to understand. Rewrite the abstract to make a sense.
Introduction
1. The introduction should be concise. No elaborate explanation of the synthetic pathway of IPP and DMAPP in the MEP pathway is needed.
2. Line 47, references are required.
3. The writing style has to change to discuss the references in several places, some examples:
Line 117: A team at Huazhong University of Science and Technology
Line 126: The only research group at the University of Massachusetts
Line 138: In the previous study of the research group
Line 151: The study group obtained
Line 154: The team also found that
4. References are needed several places in both the introduction and discussion part.
Results and discussion
Rewrite the results part to make it concise. Authors can present a maximum 7 to 8 figures in the main manuscript, others should be supplementary.
1. Figure 1 and 2 should be combined.
2. Figure 4 and 5 should be present as supplementary figures.
3. Figure 3 and 6 should be combined.
4. Figure 7, Sequence alignment and domain analysis of MYC transcription factors in Taxus, it should be present as a supplementary figure.
5. Figure 8 and 9 should be combined.
6. Figure 10, 11 and 12 should be combined.
7. Figure 12 and 19, Authors should perform the interaction experiment between MYC and MED25 in one frame.
8. Figure 13, 14 and 15 should be present as supplementary figures.
9. Figure 17 and 18 should be combined.
10. Figure 19 and 20 should be combined.
11. Figure 21: what it means dark red solid line indicates a very large upregulation, use the correct phrase and explain it.
12. Write a few sentences about the model in the result section.
13. The results published by the University of Massachusetts in 2015 show:
This type of sentences was used throughout the manuscript. It should be corrected. See the style of other manuscripts.
14. When it is transferred to the overexpressed internal reference activated by the 35S promoter, the value is far lower than the Arabidopsis protoplasts used in this study, indicating that its efficiency is not high. The gene gun method may also lead to inhibition of gene expression, so the conclusions obtained in this paper have a large discussion space: A proper justification is essential. Otherwise, results are not believable.
15. Rewrite the discussion part, use a few paragraph to explain it.
Material and methods
1. Line 613: GV3101 soil rhizobium by electroporation, is it correct?
2. A list of primers was required, which was used in race PCR and genome walking.
Author Response

(The authors gave the same response as above.)
